# Use of Digital Images as a Low-Cost System to Estimate Surface Optical Parameters in the Ocean

**DOI:** 10.3390/s23063199

**Published:** 2023-03-16

**Authors:** Alejandra Castillo-Ramírez, Eduardo Santamaría-del-Ángel, Adriana González-Silvera, Jesús Aguilar-Maldonado, Jorge Lopez-Calderon, María-Teresa Sebastiá-Frasquet

**Affiliations:** 1Facultad de Ciencias Marinas, Universidad Autónoma de Baja California, 22860 Ensenada, Baja California, Mexico; 2Institute of Water Engineering and Environment (IIAMA), Universitat Politècnica de València (UPV), Camino de Vera s/n, 46022 Valencia, Spain; 3Institut d’Investigació per a la Gestió Integrada de Zones Costaneres (IGIC), Universitat Politècnica de València (UPV), 46730 Grau de Gandia, Spain

**Keywords:** low-cost tools, coastal monitoring, marine optical properties, ocean color, digital photography, digital colors (RGB)

## Abstract

Ocean color is the result of absorption and scattering, as light interacts with the water and the optically active constituents. The measurement of ocean color changes enables monitoring of these constituents (dissolved or particulate materials). The main objective of this research is to use digital images to estimate the light attenuation coefficient ($K_{d}$), the Secchi disk depth ($Z_{SD}$), and the chlorophyll a ($Chl_{a})$ concentration and to optically classify plots of seawater using the criteria proposed by Jerlov and Forel using digital images captured at the ocean surface. The database used in this study was obtained from seven oceanographic cruises performed in oceanic and coastal areas. Three approaches were developed for each parameter: a general approach that can be applied under any optical condition, one for oceanic conditions, and another for coastal conditions. The results of the coastal approach showed higher correlations between the modeled and validation data, with $r_{p}$ values of 0.80 for $K_{d}$, 0.90 for $Z_{SD}$, 0.85 for *C**h**l_a_*, 0.73 for Jerlov, and 0.95 for Forel–Ule. The oceanic approach failed to detect significant changes in a digital photograph. The most precise results were obtained when images were captured at 45° (n = 22; $F_{r\;cal}=11.02>F_{r\;crit}=5.99$). Therefore, to ensure precise results, the angle of photography is key. This methodology can be used in citizen science programs to estimate *Z_SD_*, *K_d_*, and the Jerlov scale.

## 1. Introduction

Human activity involves direct or indirect consumption goods from ecosystems, known as ecosystem services (ES) [1,2]. Unrestrained exploitation has led to the exhaustion or collapse of some of these natural resources [3,4]. Planning rational use without compromising future use requires an understanding of ecosystemic variability [3,4,5,6]. To this end, it is necessary to define the primary variables that can describe the changes in ecosystems and define its baseline [1,7,8]. The baseline represents the combination of natural and anthropogenic variability [1,4]. 

To build the baseline, monitoring programs need to generate long enough time series. Current long-term monitoring initiatives include the Latin American Marine Monitoring Network (ANTARES) [9], the Hawaii Ocean Time Series (HOT) [10], and the California Current Ecosystem Long Term Ecological Research (CCE-LTER) [11]. However, the implementation of these monitoring initiatives involves high investment and maintenance costs [1]. These costs are particularly high in the case of marine ecosystems due to their complexity. These costs are generally incurred by government agencies, either as part of national programs (mostly) or by international conventions (less important), which implies a challenge because the budget can vary or be canceled at the end of each legislative period [1].

An alternative to overcome this challenge is to supplement monitoring programs with citizen monitoring programs, in which low-cost and easy-to-apply methodologies are used for the evaluation of key variables representing changes in the ecosystem [12]. These programs should reflect the work conducted by citizens collaborating with scientists or under their leadership [13] and represent an alternative that can help generate high-quality data with broad spatiotemporal coverage, leading to a better understanding of ecosystems [12,14,15] and, therefore, of the ES linked to them.

This work focuses on marine ecosystems for two reasons: the high cost of traditional monitoring programs and the intense anthropic pressure to which they are subjected [16,17]. Discharges in the sea produce increased levels of colored dissolved organic material (CDOM) [18,19] and particulate substances [20,21]. These compounds affect the light field in the water and the phytoplankton community (short-lived organisms that reflect short-term changes) [22,23,24]. Consequently, seawater has a characteristic color, which, together with transparency, is used to classify it in optical terms [25,26]. Changes in the optical classification of water reflect changes in the ecosystem [27].

There have been several attempts to classify water plots based on their color and transparency [28]. The first classifications were subjective and qualitative [29]. The classification proposed by Pietro Angelo Secchi [30] was based on estimating water transparency according to Secchi disk depth ($Z_{SD}$), which is the depth at which a Secchi disk is no longer viewable by an observer when it is lowered into the water. $Z_{SD}$ is a visual turbidity assessment that is inversely proportional to the amount of attenuating material present in the water column. Although it represents a quantitative measure of the transparency of a water body, it is considered subjective and qualitative because it relies on the human eye and, therefore, different people can record different disk readings. The Forel–Ule (FU) scale [31,32] is a sea color comparator scale that was developed to cover all possible natural sea colors. It consists of vials with fluid of 21 colors ranging from blue to brown; water samples are classified by matching colors. The development of underwater radiometric equipment allowed for quantitative classification schemes to relate optical parameters to the observed variability of water transparency and color [29]. The Jerlov scale [33] established five oceanic water types (I, IA, IB, II, and III) and five coastal water types (1, 3, 5, 7, and 9) based on measurements of the light attenuation coefficient ($K_{d}$) (a parameter that describes the attenuation of light in the water column). The classification of Morel and Prieur [34] sorts seawater into two types—case 1 and optically complex waters—based on their reflectance and light absorption coefficient (*a(λ)*). In recent studies, optical classifications have also been made based on products obtained from remote sensing, such as Moore’s optical water types, which are based on remote sensing reflectance (Rrs) [35,36,37,38,39,40,41,42,43,44,45]. For instance, the FU scale is applicable to remote sensing data thanks to new algorithms that convert remote sensing reflectance (Rrs) from satellite-borne ocean color sensors to FU [46]. 

The ability to obtain RGB digital color intensities from digital camera images taken from fixed platforms, boats, or unmanned aerial vehicles has been evaluated in other studies [47,48,49,50,51,52,53,54]. The objective of these studies was to obtain water-leaving radiance [54] or RGB reflectance to estimate surface optical parameters [48,49,50,51,53]. These latter approaches are based on the mixing ratio of each color, ranging from 0 to 255, where 0 indicates the absence of color and 255 indicates the maximum mixing ratio. Accordingly, each color is defined according to three numbers or digital values (R, G, B) [48,49,50,51,53,54]. Once digital values have been obtained, they are associated with in situ optical parameters of the studied surface [48,49,50,51,53,54].

For marine waters, Goddijn-Murphy et al. [48] developed two approaches to estimate surface chlorophyll a $\left(Chl_{a}\right)$. These authors used conventional digital cameras, the Nikon Coolpix885(®) (Minato, Tokyo, Japan) and the SeaLife ECOshot(®) (Moorestown, NJ, USA), to perform measurements at 12 stations located in Galway Bay (North Atlantic). They used a Nikon Coolpix885 (®) with a plastic tube fitted around the lens to break the air–water interface to prevent surface-reflected light from entering the camera and to allow the camera to capture only the water-leaving radiance. They also used a SeaLife ECOshot(®), a waterproof digital camera, to operate below the water surface, taking pictures of upwelling light; they then inverted the setup it to capture downwelling sunlight. In this work, they obtained a significant but low relationship between colors B and G colors of the images and the concentration of $Chl_{a}$. 

Based on the previous results and with the development of smartphones, the cameras integrated into these devices began to be used to remotely monitor the optical characteristics of a water parcels using mobile applications (app), such as: The Secchi3000 app 2.0, which was designed to estimate turbidity and Z_SD_. Along with the app, a simple and cheap device is provided that must be filled with the water to be tested; then, a photograph must be taken in specific areas of the sample. The image is sent to the server, where it is processed using pattern recognition and computer vision techniques [49];TheEyeofWater app 2.4.0, which allows for evaluation of water color. The app guides the user to capture an image of the water surface. After capturing the image, the observed water is assigned one of the 21 colors on the Forel–Ule scale [50]The HydroColor app 2.0, which allows images of the water surface to be taken following a specific protocol that includes the positioning angle of the camera and the observer. Subsequently, the app processes the colors of the image to estimate the turbidity of the water [51].

The aim of this research is to increase the possibly of using digital camera images to determine key optical parameters, specifically to estimate $K_{d}$ and $Z_{SD}$. $K_{d}$ is an indicator of the turbidity of the water column, an apparent optical property (AOP) that is a property of water that changes with a changing light field [27]. It is directly related to the concentration of scattering particles in the water column; non-algal particles, phytoplankton, CDOM, and water itself are considered the four optically significant substances that control it [27]. It is an important parameter for water quality that can be used to predict the euphotic depth and estimate primary productivity; it is also essential for monitoring of the eutrophication process due to light attenuation by phytoplankton growth or suspended matter [55,56]. It is important to maximize the benefits of this low-cost methodology.

The objectives of the present work are to (1) use digital images to estimate the optical parameters $K_{d}$, *Z_SD_*, and *C**h**l_a_* and to optically classify plots of seawater according to the Jerlov scale and the Forel–Ule scale; (2) evaluate the effect of camera angles relative to sea surface when capturing such images; and (3) describe the validated image-capturing methodology to be used in citizen science programs.

## 2. Materials and Methods

The database used in this study was obtained from seven oceanographic cruises performed in oceanic and coastal areas Figure 1. The cruises Pangas 0613, Vaquita 0716, and Exfinife 0916 were conducted in the Gulf of California (Pacific Ocean, Mexico). The cruises Point Sur 0413, Glyders 0615, Marias 0316, and Marias 0916 were conducted in the Northeastern tropical Mexican Pacific Figure 1. During these cruises, digital images of the sea surface were captured, in addition to measurements of light irradiance in the water column (*E_d_* (*PAR*)), *a(λ)*, *Z_SD_*, and *C**h**l_a_* on the surface, using the methodologies described below. The number of samples obtained for each variable in each cruise is shown in Table 1.

The digital images were captured on an iPad Air 2 with an 8 megapixel camera (San Diego, CA, USA). The Spyglass application [57] was downloaded on this device to determine the tilt angle of the camera and capture the images. The locations of stations were determined by integrating a Bad Elf GPS with the device. 

The images were captured following the protocol of Deschamps et al. [58] on days with nil or little cloud cover (maximum 30% coverage allowed) and at a time of the day when the sun was more than 45° above the horizon (between 10:00 am and 4:00 pm for mid-latitudes). To minimize residual polarization and quantify the emerging radiance of water, the observer applied the following steps [58,59]: Position his/her body at the bow of the vessel with the sun on his/her back (the sun must be at an angle greater than or equal to 45° relative to the horizon). It is recommended to be located at the bow of the vessel because it is the narrowest and least shady area Figure 2a;Rotate the body 45° to the right or left of the starting position and select a position where the shadow of the vessel is not projected upon the photographic field. According to the methodology proposed by Deschamps et al. [58], this position is required because the observer must be positioned at 135° in the azimuth between the position of the sun and his/her visual field to facilitate measurements from any platform in the ocean, including moving vessels Figure 2b;Place the camera at 45°relative to the sea surface, and capture the image Figure 2c.

At least six digital images were captured at each station. We selected images captured at exactly 45° from the sea surface and with no foam within the central zone. To obtain the digital colors (R, G, B), the images were processed in Corel Photo Paint X8. According to Leeuw and Boss [51], for each image, four 1 cm × 1 cm quadrants were selected; in each quadrant, a color histogram was obtained, from which the digital values (R, G, B) were extracted Figure 3. To reduce the natural variability of water color, the digital values of each station were taken as the mean of the values obtained from all quadrants in all the images selected for each station.

To estimate the Forel–Ule scale for each station, we applied the criteria of Novoa et al. [60], and the digital values (R, G, B) were converted to chromaticity coordinates (*x, y, z*) according to the definition of the International Commission on Illumination—*x+ y + z* = 1; therefore, *z* = 1 − *x – y*; hence, the coordinate (*z*) provides no additional information, so only the coordinates (*x, y*) are used to represent a color in a chromaticity diagram [60,61]. 

Once the chromaticity coordinates (*x, y*) for each image were obtained, they were contrasted with the coordinates estimated by Novoa et al. [60] for each type of Forel–Ule water Table 2. This was carried out by applying the least-squares criteria based on a goodness-of-fit test [62,63].

Measurements of $Z_{SD}$ were carried out simultaneously when capturing images of the sea surface. A 30 cm diameter oceanographic disk was used for $Z_{SD}$ measurements, which was lowered into the water column from the sunny side of the vessel [26]. The depth at which the disk disappeared from the observer’s view (to the naked eye) was recorded as $Z_{SD}$ [30,64]. The *E_d_* (*PAR*) measurements were conducted with a Li-Cor scalar irradiometer (LI-193) (Lincoln, NE, USA), which recorded the light at 1 m intervals across the water column up to a maximum depth of 30 m.

Subsequently, *K_d_* was estimated based on two methodologies. In the first methodology, it was calculated indirectly from the measurements of $Z_{SD}$ and by applying the criteria of Castillo-Ramirez et al. [56]. In the second methodology, the measurements of *E_d_* (*PAR*) were used based on the criteria of Kirk [27], as expressed in the following equation (Equation (1)):(1)$$ln\left(PAR\left(z\right)\right)=ln\left(PAR\left(0\right)\right)-K_{d}\times Z\;$$
where *K_d_* is the slope of a linear regression, and the dependent variable is the natural logarithm of irradiance as a function of depth.

Likewise, the oceanographic cruises collected water samples to estimate *a(λ)* based on the criteria of Mitchell et al. [65] and the concentration of *C**h**l_a_* following the methodology of Thomas [66]. *a(λ)* was estimated considering the light absorption of pure water (*a_w_*(*λ*)), phytoplankton (*a**_phy_*(*λ*)), and CDOM ((*a**_CDOM_* (*λ*)) [67] (Equation (2)).
(2)$$a\left(\lambda \right)=a_{w}\left(\lambda \right)+a_{phy}\left(\lambda \right)+a_{CDOM}\left(\lambda \right)$$

The estimated *a(λ)* was used to classify the stations according to the Jerlov optical water types following the criteria of Solonenko and Mobley [67] and Castillo-Ramirez et al. [56].

The database described above covers a range of optical conditions, from clear (oceanic) to turbid (coastal) waters (Figure 1). For this reason, three types of empirical approaches were generated for each optical variable (Forel–Ule scale, *Z_SD_*, *K_d_*, Jerlov scale, and *C**h**l_a_*): a general approach that can be applied under oceanic and coastal conditions and two specific approaches to be applied separately under these conditions (oceanic and coastal approaches) Figure 4. The full database, including oceanic and coastal waters, was used for the general approach. For the oceanic and coastal approaches, the stations were classified first by the Jerlov scale Figure 4.

The empirical approach relates the Forel–Ule scale, *Z_SD_*, *K_d_*, the Jerlov scale, and *C**h**l_a_* with the digital values (R, G, B) based on a stepwise multiple regression analysis following the Bass criteria [68] (Equation (3)).
(3)$$\hat{y}=b_{o}+b_{1}R_{1}+b_{2}G_{2}+b_{3}B_{3}$$
where $\hat{y}$ is the variable to predict, which can be the Forel–Ule scale, $Z_{SD}$, $K_{d}$, Jerlov scale, a $Chl_{a}$ (dependent variable), with (R, G, B) as the independent variables and $b_{n}$ as their associated coefficients.

On the other hand, to eliminate unusual observations, an analysis of residuals was applied based on the following equation (Equation (4)):(4)$$e=y-\hat{y}$$
where e is the residual; it can be considered as the error calculated as the distance divided by the observed value (y) and the modeled value ($\hat{y})$. Residuals closer to zero indicate a better model performance.

To identify high-noise observations, the criteria of the Six Sigma analysis were followed based on standardized errors [64], where *e* (*Z**e*) is standardized (Equation (5)).
(5)$$Z_{e}=\frac{e-\bar{e}}{SD_{e}}$$
where $\bar{e}$ is the average of the residuals, and $SD_{e}$ is the standard deviation of the residual.

Once the high-noise observations were removed, the database for each variable was randomly split into two sets: 50% for modeling and 50% for validating purposes [69,70]. To reduce the random error in the selection of the two datasets and test the robustness of the models, we performed five iterations by randomly selecting five different datasets to model, with their respective validation set (Figure 4). A stepwise multiple regression was applied to the data used for modeling, following the criteria of Bass [68].

Subsequently, to demonstrate that the independent variables were significant in each model, a *t*-test (Equation (6)) was applied to the coefficients associated with each of these variables.
(6)$$t_{cal}=\frac{b_{k}}{SE_{bk}}$$
where $b_{k}$ is the regression coefficient associated with the independent variable, and $SE_{bk}$ is the standard error of the coefficient ($b_{k}$) expressed in the following equations (Equation (7)–(9)):(7)$$SE_{b1}=\sqrt{\frac{MSE}{\sum R^{2}-{\frac{\left(\sum R\right)}{n}}^{2}}}$$
(8)$$SE_{b2}=\sqrt{\frac{MSE}{\sum G^{2}-{\frac{\left(\sum G\right)}{n}}^{2}}}$$
(9)$$SE_{b3}=\sqrt{\frac{MSE}{\sum B^{2}-{\frac{\left(\sum B\right)}{n}}^{2}}}$$
where (R, G, B) are the digital colors (independent variables).

To test the general significance of the resulting models, an *F*-test was run based on Equation (10): (10)$$F_{cal}=\frac{\frac{\sum {\left(y_{i}-\bar{y}\right)}^{2}-\sum {\left(y_{i}-\hat{y}\right)}^{2}}{k}}{\frac{\sum {\left(y_{i}-\hat{y}\right)}^{2}}{n-\left(k+1\right)}}=\frac{MSR}{MSE}$$
where k is the number of independent variables (three in this case), MSR is the squared mean of the regression, and MSE is the squared mean of the residual error.

The proportion of the variation of the dependent variable that can be explained by independent variables was estimated with the coefficient of determination ($R^{2}$) (Equation (11)).
(11)$$R^{2}=\left(\frac{\;\sum {\left(y_{i}-\hat{y}\right)}^{2}}{\sum {\left(y_{i}-\bar{y}\right)}^{2}}\right)\times 100=\left(\frac{SSE}{SST}\right)\times 100$$
where SSE is the variability explained by the model, and SST  is the variability explained by y.

Once the models for each variable were obtained, they were validated based on a match-up analysis [71], where the statistical validity of the models was estimated using the Pearson correlation coefficient (*r**_P_*), expressed as (Equation (12)):(12)$$r_{P}=\frac{Cov_{model,val}}{{\mathrm{SD}}_{model}\times SD_{val}}$$
where *r*_*P*_ is the Pearson correlation, *Cov**_model,val_* is the covariance of the modeled and validated datasets, and *SD_model_* and *SD_val_* are the standard deviations of the modeled and validated datasets, respectively. This coefficient is a measure of the linear correlation between two variables; it ranges between −1 and +1 (where +1 indicates a direct linear relationship, −1 indicates an inverse linear relationship, and 0 indicates a nonlinear relationship).

In order to compare the models estimated in this work with those from the literature, three statistical descriptors were calculated: mean absolute error (MAE) (Equation (13)), root-mean square error (RMSE) (Equation (14)), and analysis of bias (*BIAS*) (Equation (15)).
(13)$$MAE=\frac{\sum \left|\left(K_{d_{in\;situ}}-K_{d_{model}}\right)\right|}{n}$$
(14)$$RMSE=\sqrt{\frac{\sum {\left(K_{d_{in\;situ}}-K_{d_{model}}\right)}^{2}}{n}}$$
where *n* is the total number of data points included in this analysis, $K_{d_{in\;situ}}-K_{d_{model}}$ represents residual observations, and $\left|K_{d_{in\;situ}}-K_{d_{model}}\right|$ is the absolute value of residuals.
(15)$$BIAS=average\;\left(K_{d_{in\;situ}}-K_{d_{model}}\right)$$
where *BIAS* is the residual mean.

Lower MAE and RMSE values represent better results, whereas *BIAS* values closer to zero indicate better results. To determine which was the best model, the model performance index (MPI) was estimated [56] (Equation (16)), which is based on the three statistical descriptors mentioned above.
(16)$$\;MPI=1-\left(\frac{\left(\frac{R_{MAE}}{p}\right)+\left(\frac{R_{RMSE}}{p}\right)+\left(\frac{R_{\left|BIAS\right|}}{p}\right)}{3}\right)$$
where $R_{MAE},\;R_{RMSE\;}$ are the range of MAE and RMSE, respectively; $R_{\left|BIAS\right|}$ is the absolute range of BIAS; and *p* is the total number of compared models. The ranks were calculated following the criteria of Wilcoxon [72]. The MPI intervals range from 0 to 1, where values closer to 1 represent a better model. 

To evaluate whether the differences between the in situ values and the results obtained from the 45° and non-45° images were statistically significant, the models mentioned above were applied for those stations that met the following criteria: (1) stations that had in situ data for the variable to be modeled, (2) R, G, B values obtained from images captured at 45°, and (3) R, G, B values obtained from images captured at an angle other than 45°. Following in situ quality control, whereby only images captured at or close to a 45° angle were retained, these data only included images captured with a ± 1° difference (44° and 46°). A non-parametric ANOVA analysis was performed by Friedman blocks following the criteria of Friedman [73] (Equation (17)).
(17)$$F_{r\;cal}=(\left(\frac{12}{rk\left(k+1\right)}\times \sum (T_{k}^{2}\right)))-3r\left(k+1\right)$$
where *r* is the number of observations, *k* is the number of treatments, and $T_{k}$ is the sum of the ranges in each treatment.

Once the differences were analyzed and to determine which results presented the lowest error, a standardized residual analysis was run (Equation (5)). Values closer to zero indicates a lower error.

To estimate the accuracy of the results, a confidence interval was established based on *t_crít α/2, n−1_* with an α of 95%; then, we calculated the percentage of the data that were within this interval. Finally, the data accuracy was tested based on a least-square test applied to the residuals following the criteria of Xu et al. [62]; the most accurate data are those with the lowest *X*^2^ value.

## 3. Results and Discussion

The approaches proposed in this work are empirical; therefore, they depend on the boundary limits established by the variables used in their development [26,74]. Empirical modeling of ocean color and optical parameters is challenged by limited sampling opportunity on sunny days with calm seas. These conditions are essential to obtain data that represent the true variability of the light field in water [75,76,77]. To address this challenge, in this research, we used data from seven oceanographic cruises carried out between 2013 and 2016. In each cruise, a sampling network of more than 70 stations was established. However, data could only be obtained at approximately 15% of the established stations due to unfavorable conditions (e.g., cloudy days and waves) Table 1. In addition, it is necessary to obtain an independent dataset to validate the developed model. This validation is essential to assess the ability of the model to predict values [26,74]. 

To address these challenges, the database for each variable was divided into two independent groups, using 50% of the data for the modeling process and the other 50% for validation. In addition, to ensure the robustness of the models, five iterations were performed randomly, selecting five different datasets to model, each with its corresponding validation set [56,69,70] (Figure 4). This approach helped to reduce the random error associated with the selection of the two datasets.

### 3.1. General Approach

The general approach was estimated from the complete database with optical conditions ranging from oceanic to coastal. This showed that the non-significant independent variable for the estimation of most parameters was the digital color R corresponding to the red wavelength (700 nm) Table 3. This is because the light absorption by water increases exponentially toward the red region of the electromagnetic spectrum [78], which implies a lower penetration capacity into the water column, a lower reflection of long wavelengths (~700 nm), and, therefore, fewer possibilities to react to changes in optically active compounds [27,79]. The only case in which the three digital colors (R, G, B) were significant Table 3 was to estimate the Forel–Ule scale (Equation (18)). This result may have occurred because this scale, unlike the parameters used in the other approaches, is based on a visual perception of water color, which results from the combination of these three colors (R, G, B) [80].

Once the significant variables were identified, we confirmed that the general approach for all the parameters (Equations (18)–(22)) was significant (Table 3, column 10 ($F_{cal}$*>*$F_{cri}$)). This finding implies that the partition of a digital image into digital channels (R, G, B) and the association of these channels with in situ data of surface optical parameters can be also used to estimate *Z_SD_*, *K_d_*, and the Jerlov scale. In addition, these findings support the reports by Goddijn-Murphy et al. [48], Novoa et al. [50], and Leeuw and Boss [51], who suggested the use of images as an alternative to estimate *C**h**l_a_*, turbidity, and the Forel–Ule scale.

Subsequently, this approach was validated based on the criteria proposed by Gregg and Casey [81], Djavidnia et al. [82], and Santamaría-del-Ángel et al. [71], who established that *r_pcal_* values above 0.70 indicate a strong association. These criteria suggest that the correlations between the modeled and validation data were mostly highly significant Table 4. The *K_d_* model expressed by (Equation (20)) yielded the lowest values. To improve *K_d_* estimation, the criteria of Castillo-Ramirez et al. [56] were used based on values modeled according to (Equation (19)). The Castillo-Ramirez et al. [56] criteria yielded higher correlation values ($r_{pcal}$= 0.85) Table 4; therefore, we propose this as the best alternative for estimating *K_d_* from a digital image. 

The results presented in Table 3 and Table 4 show that the general approach can be applied for a broad spectrum of optical conditions to estimate all the modeled parameters ($Z_{SD}$, *K_d_*, Jerlov scale, *C**h**l_a_*, and the Forel–Ule scale). 

### 3.2. Oceanic and Coastal Approaches

Morel and Prieur [34] proposed the classification of seawater into two types: case 1 and optically complex waters. Case 1 waters are those whose optical properties are driven mainly by phytoplankton and are generally found in oceanic areas far from the continental shelf. Optically complex waters contain suspended sediments, non-phytoplanktonic organic particles, or CDOM, in addition to phytoplankton. The sources of these compounds are frequently associated with coastal areas; however, in some cases, it is also possible to observe optically complex waters in oceanic areas. For instance, a phytoplanktonic bloom would increase CDOM levels in an oceanic water plot and would provide it with optically complex characteristics [34,79,83].

The approaches for oceanic and coastal conditions were obtained to assess whether more specific models (in terms of optical conditions) are more accurate to estimate the studied parameters. The results of the oceanic approach Table 5 show that none of the three digital colors (R, G, B) was significant for most of the parameters. The only significant model for the oceanic approach was that for the estimation of the Forel–Ule scale (Table 5; Equation (23)). The greatest quantity of optical components that produce a change in water color occurs in coastal areas, where they have a high variability [34]. The results obtained in the present work Table 5 indicate that water color changes in oceanic regions are not significant enough to be captured in a digital image to be associated with *Z_SD_*, *K_d_*, the Jerlov scale, and *C**h**l_a_*.

The coastal approach showed that the digital color G was significant in all models, for some parameters in combination with the digital colors R or B Table 6. This variation in significant digital colors (or wavelengths) depends on the nature and quantity of particles present in the studied water plot [25]. Specifically:Phytoplankton produces a green coloration in the water when it is in high concentrations due to the presence of *C**h**l_a_* in cells, except for certain species that produce a red or brown coloration;Non-phytoplankton material (or detritus) produces a brown or reddish coloration depending on the source of the material;CDOM stains water a yellow–brown color.

The coastal approach was significant to estimate all the variables (Equations (24)–(28)) Table 6, the Forel–Ule scale, the Jerlov scale, *Z_SD_*, *K_d_*, and *C**h**l_a_* under optically complex conditions. The degree of association between modeled and validation data (*r_Pcal_*) for the oceanic and coastal approach Table 7 was higher than that obtained in the general approach Table 4. The validation of these models shows that a more specific approach in terms of optical conditions results in greater accuracy Table 7. The model that yielded lower values than those reported for the general approach was that predicting the Jerlov scale (Equation (27)). This may be because the model proposed by Solonenko and Mobley [67] for associating *a(λ)* with the Jerlov water type ignores the contribution of non-phytoplanktonic particulate matter. This component includes phytoplankton, detritus, and other organic particles and minerals (Equation (2)), which represent important contributions in coastal areas, as reported by Morel and Prieur [34].

### 3.3. Model Comparison

Table 8 shows a comparison of the performance of our $Chl_{a}$ general and coastal models (Equations (22) and (28)) with models proposed by Goddijn-Murphy et al. [48]. These authors developed two models, in which the B/G ratios are the independent variables and $Chl_{a}$ is the dependent variable.

The results presented in Table 8 show that the models proposed in this work achieved a better performance in the validation process compared to those reported in the literature applied to our data, demonstrating the relevance of acquiring field data under appropriate conditions (e.g., sunny days and calm sea). These conditions were not observed by Goddijn-Murphy et al. [48]. Additionally, the lower R^2^ obtained by Goddijn-Murphy et al. [48] could be due to the high CDOM concentrations reported, which could affect the absorption of blue wavelengths in water. 

Gao et al. [53] developed an algorithm to estimate $Z_{SD}$ using smartphone images in continental water bodies. These authors observed the same conditions as in our study, including camera and sun angles. However, we could not apply their model to our data because it was developed for other conditions. Continental waters with high CDOM concentrations are characterized by high reflectance in R and high absorption in B [48]. They also used a limnologic Secchi disk, while in this research, an oceanic disk was used. Differences in terms of size and reflectance surface are noticeable in both versions [26,56], so they are not comparable.

### 3.4. Angle Effect

Friedman’s non-parametric analysis showed significant differences between the *Z_SD_* (Equation (25)) and the *C**h**l_a_* (Equation (28)) values obtained from an image captured at 45° and at an angle other than 45° Table 9. 

Subsequently, the precision of the results was evaluated; to this end, a confidence interval of ±2.08 was set when using the model to predict *Z_SD_* (Equation (25)) and ±2.06 for *C**h**l_a_* (Equation (28)) Figure 5a,b. This methodology showed that when applying the *Z_SD_* model (Equation (25)), 95.45% of the data obtained from 45° images were within the confidence interval, while only 86.36% were within this interval for non-45° images Figure 5a. It was also observed that the data outside the confidence interval were underestimated. On the other hand, when we applied the *C**h**l_a_* model (Equation (28)), 95.45% of the 45° data and 90.90% of non-45° data fell within the confidence interval Figure 5b. In this case, the data outside the interval are overestimated. Once the differences and precision of the results were analyzed, their accuracy was evaluated; the lowest *X*
^2^ values correspond to the results obtained using 45° images Table 10. This finding indicates that these results, in addition to being more precise, are more accurate.

The assessment of the variability of images according to the angle at which they were captured Table 9 and Table 10 and Figure 5 showed that a variation of just one degree in the position relative to the surface may lead to significant differences in the results.

## 4. Discussion

The methods traditionally used for in situ monitoring to estimate optical parameters involve the use of specialized instrumentation, which can be relatively complex and expensive [63,66,67,84,85,86,87,88,89,90]. In addition, these methods involve a slow analytical process, making them ineffective to obtain a large-scale view of the study area in a short time [48,90]. On the other hand, the logistics involved in preparing in situ monitoring based on optical parameters is not straightforward; it requires a vessel, field material, and personnel experienced in recording measurements and collecting samples. These monitoring procedures should be scheduled on sunny days, as clouds can distort light measurements in the water during field work [75,76,77].

The results of this study show the potential of digital images to evaluate the surface optical parameters of a water plot. The main advantage of the proposed approaches is their easy implementation and low cost, since they do not demand optics expertise and only require a digital camera. These advantages can be summarized as follows [47,48,49,50,51,52,53,54,91,92,93,94,95,96]:Low-cost system: The use of smartphones, tablets, or digital cameras to capture digital images is a cost-effective method for estimating surface optical parameters in the ocean, as it does not require expensive specialized equipment. Smartphones and other devices are relatively inexpensive and widely available, making them an attractive option for researchers who are working with limited budgets or who do not have access to specialized instrumentation;Accessibility and ease of use: since these electronic devices are widely used, this methodology can provide a widespread network of data acquisition, which can be essential for global-scale analyses and modeling;High spatial and temporal resolution: digital image capture can provide high spatial and temporal resolution, allowing for a more detailed analysis of the ocean’s surface optical properties over a larger area;Versatility: smartphones, tablets, or digital cameras can be used to capture images from different platforms such as beaches, piers, and boats, making them versatile tools for surface optical parameter estimation;Citizen science: The use of digital images allows for citizen scientists to contribute to the data acquisition process. This can increase public participation in scientific research and environmental monitoring, as well as their awareness and engagement in oceanographic research;Environmental monitoring: Surface optical parameters play a crucial role in the health of marine ecosystems. The use of electronic devices can aid in environmental monitoring efforts, providing critical information for decision-making and conservation efforts;Rapid response: in the event of a phytoplankton bloom, the use of digital images can aid in rapid response efforts, providing real-time information on the extent and severity of spills.

This type of low-cost, user-friendly approach not only benefits scientists but could also be used by the tourism or aquaculture industries, as well as by individual citizens concerned about changes in the water quality in their local environment. The benefits of this approach go beyond just improving data collection. By enabling real-time and continuous monitoring of surface optical parameters, this approach provides valuable information for environmental management and decision making. The enhanced resolution and coverage of the data also offers new opportunities for researchers to investigate the complex relationships between surface optical parameters and environmental processes. This is especially important in coastal areas, where our results are more accurate, since 355 million people are expected to live within the 100 km of the coast between 2020 and 2035 [16,97,98]. This continuously increasing pressure will likely lead to the degradation of these ecosystems, adversely impacting the provided ES [26,99]. We propose implementing the use of the coastal approach to supplement traditionally used analyses for in situ monitoring.

Taking into account these advantages, using the model generated for the Jerlov scale (Equation (27)), as an example, would help to classify water types in a quick and efficient way, since the equipment currently used to define water types based on this scale require specialized knowledge for their use. This is the case of hyperspectral irradiometers [87] or spectrophotometers (for calculating inherent optical properties) if the criteria of Solonenko and Mobley [67] are followed, as in the cruises used to build the database used in the present study. Likewise, approaches such as that proposed by Mallick et al. [89], where *K_d_* values are derived from satellite images, may be considered for estimating the Jerlov scale. However, as mentioned by Lebourgeois et al. [47], the processing of such images is more complex relative to digital images. In turn, the approach proposed herein could be used to obtain and monitor optical properties such as $a\left(\lambda \right)$, the total dispersion coefficient ($b\left(\lambda \right)$), and *K*_d_ $\left(\lambda \right)$, since, as reported by Jerlov [33] and Solonenko and Mobley [67], each Jerlov water type is associated with a typical spectrum of these properties. Therefore, a shift in the Jerlov water type may indicate that the components in the water column (phytoplankton, detritus, and CDOM) and the light field are changing. 

Estimating *C**h**l_a_* concentrations with the model proposed in this work (Equation (28)) would help us to monitor phytoplankton blooms [100,101] at a lower cost and more quickly. This is because the methodologies currently used in laboratories involve equipment such as spectrophotometers or high-performance liquid chromatography (HPLC), which require a filtration system, fiberglass filters, and solvents [66,102]. In addition, the methodology involving HPLC also requires standards of pigments to estimate their concentration and takes approximately 48 h [66]. However, phytoplankton blooms are proliferation events that can last less than 24 h (fast blooms), for several days, or for weeks [103]. Therefore, a rapid response can be crucial. 

The estimated models to predict *Z_SD_* (Equation (25)) and water color based on the Forel–Ule scale (Equations (23) and (24)) could be used to monitor eutrophication and anoxic–hypoxic events, which influence water color and transparency [104,105,106]. In addition, they would assist us to obtain data on these variables on sunny days when a Secchi disk or color comparators are unavailable. In addition, in the case of the Forel–Ule scale, this approach allows for estimation of all 21 colors of the scale, as the currently used instrument only shows 16 color comparators. 

Thus, this approach may facilitate surveying the water status in the study area without the need to collect water samples or previous planning. For example, data could be obtained on sunny days when no field trips are scheduled. However, it is worth noting that the reliability and usefulness of this approach requires additional in situ measurements to carry out additional calibration and validation studies.

In addition, the algorithms proposed herein may be implemented in the development of an app that provides the Forel–Ule scale, *Z_SD_*, *K_d_*, the Jerlov scale, and *C**h**l_a_*. However, to obtain high-quality data, it would be essential to implement quality flags such as those used in the “Eye on water” app 2.4.0 [107], where the user is asked to perform a test prior to capturing the image.

Although the use of digital images to estimate surface optical parameters in the ocean has notable advantages in terms of ease of use and low cost, it is important to mention that there are limitations that must be considered. One issue that could arise is whether different technologies used in smartphone cameras could generate different RGB readings. However, Leeuw and Boss [51] evaluated the spectral sensitivity of RGB channels in different next-generation devices and showed that although there may be differences in the spectral shape, the values in RGB peaks are practically the same between devices. The approaches presented in this work are based on the RGB peaks Figure 3, so they should be valid regardless of the device being used. Another issue is the interpretation of color changes in a relatively small area (1 cm × 1 cm). Our methodology is based on the work of Leeuw and Boss [51], who proposed establishing a fixed area within the photograph so that devices with cameras of different resolutions can be compared with each other, thanks to the fact that the field of view of the camera is the same between devices.

Variability in lighting and image quality can be critical factors that influence the quality of the obtained data, as mentioned in [75,76,77]. The quantity and quality of light reaching the ocean surface depends mainly on the position of the sun and cloud cover [27,28]. The image quality can be affected by elements influencing the visual state of the ocean surface, for example, the presence of waves or white caps, sunshine, or the shadows generated by the boat or platform from which the image is captured [50,51,53,58,59]. 

To overcome these limitations and obtain quality data, we followed the reflectance measurement methodology with a SIMBAD spectroradiometer to capture the photographs (refer to [58]). As mentioned by Fougnie et al. [108] and Deschamps et al. [58], this methodology is very specific with respect to the sun position and angle, as well as the equipment angles, and it allows for a reduction in the noise or interference caused by the sunshine and the reflection of the sky on the water surface. Digital images were captured at an angle of 45° relative to the ocean surface on sunny days, when the sun was at an angle equal to or greater than 45° relative to the horizon and with little or no cloud cover. A 45° sun position at the zenith minimizes Fresnel reflectance on the water surface, allowing greater light penetration, improving the accuracy of radiometric measurements, and facilitating accurate estimation of optical parameters in the ocean [108].

## 5. Conclusions

Changes in color or turbidity in the marine environment involve an alteration of the components that absorb or disperse light within the water column. These changes can be associated with natural or anthropogenic processes that affect water quality, such as anoxia–hypoxia, eutrophication, and phytoplanktonic blooms. Therefore, it is advisable to implement monitoring systems to generate time series with a sufficient time span to differentiate between natural variabilities and those derived from anthropogenic pressures. 

A simple way to carry out this sort of monitoring without incurring high costs is to apply techniques such as the one proposed in the present work, which only require a smartphone or tablet. The results of this study demonstrate that images captured at the sea surface with the methodology described herein provide information about the optical characteristics (such as the Forel–Ule scale, *Z_SD_*, *K_d_*, the Jerlov scale, and *C**h**l_a_*) of a water plot. The use of these images for the development of empirical approaches yielded the best results in the coastal area. In addition, this study confirmed that if the established methodology for image capture is not followed in relation to the camera positioning angle, the results can be biased by the noise generated by the solar brightness and the reflection of the sky on the water surface. The conditions required for the proper performance of our approach that the image of the water surface be captured on days with low or no cloud cover, with a calm sea, and when the sun is behind the observer at an angle greater than or equal to 45° with respect to the horizon. Once the above conditions are met, the observer must turn his/her body 45° to the right or left of the starting position and select a position where the shadow of the boat does not interfere in the image recording field. Finally, the camera should be positioned at 45° relative to the sea surface to capture the image. 

This user-friendly and low-cost methodology could be used as a supplement to the analyses traditionally applied in in situ monitoring. The observer does not necessarily have to be an expert in optics to use it, which facilitates its implementation in monitoring schemes involving citizen assistance for data generation. This advantage would help to expand the spatiotemporal coverage of monitoring. However, it is worth noting that the reliability and usefulness of this approach must be supplemented by in situ measurements to carry out additional calibration and validation studies. Moreover, this work establishes a basis for the future development of an app to deliver the Forel–Ule scale, *Z_SD_*, *K_d_*, the Jerlov scale, and *C**h**l_a_*, which could be suitable for use by scientists and the general public. To obtain data reflecting the true variability of a water plot and filter out images that fail to meet the methodological criteria, this app should include quality flags considering the angles of the observer and the camera relative to the position of the sun. In addition, this app should record information including the date, time, global position, and distance from the water surface. The above can be achieved with the tools currently included in mobile devices, such as calendar, clock, global positioning system (GPS), and barometer. For free access to the data, we also recommend developing a web page where these data can be viewed in real time and downloaded.

## Figures and Tables

**Figure 1 sensors-23-03199-f001:**
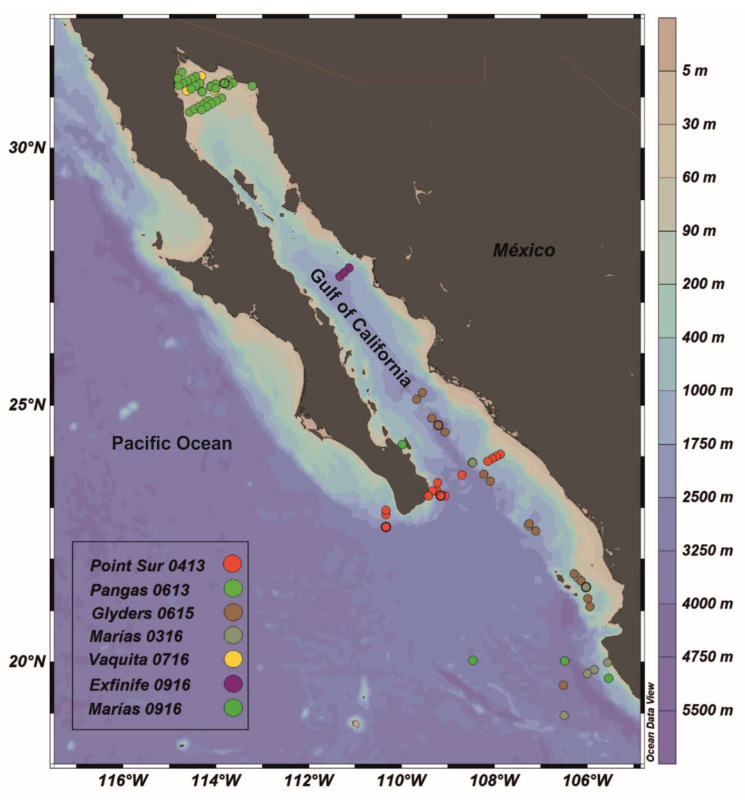
Map showing the locations of the stations used in this study.

**Figure 2 sensors-23-03199-f002:**
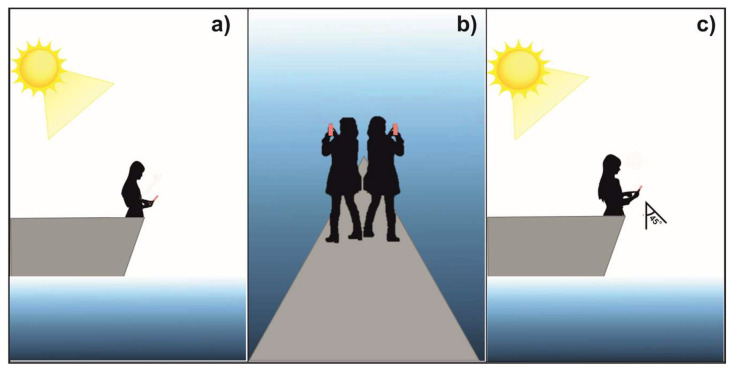
Scheme depicting the steps to follow to capture an image of the sea surface. With the sun on the back (**a**), rotate 45° to the left or right of the initial position to achieve 135° in the azimuth between the position of the sun and his/her visual field (**b**), place the camera lens at 45° relative to the sea surface and capture the image (**c**).

**Figure 3 sensors-23-03199-f003:**
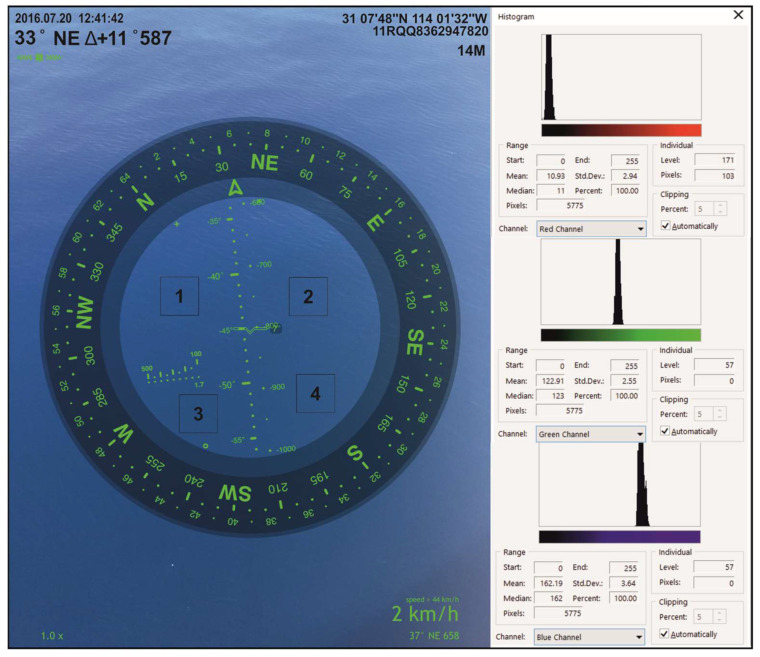
Example of digital photo processing. The four quadrants selected and the histograms used from this image to obtain the average digital values (R, G, B) are shown.

**Figure 4 sensors-23-03199-f004:**
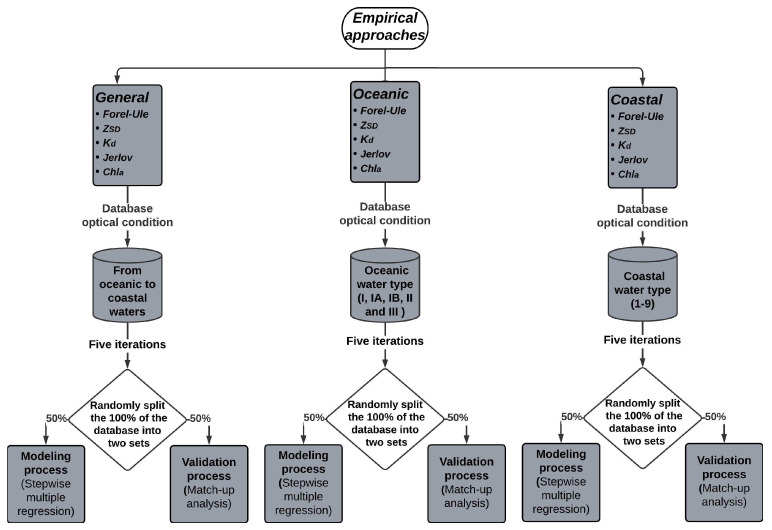
Scheme of approaches for each optical parameter and the database used for its development.

**Figure 5 sensors-23-03199-f005:**
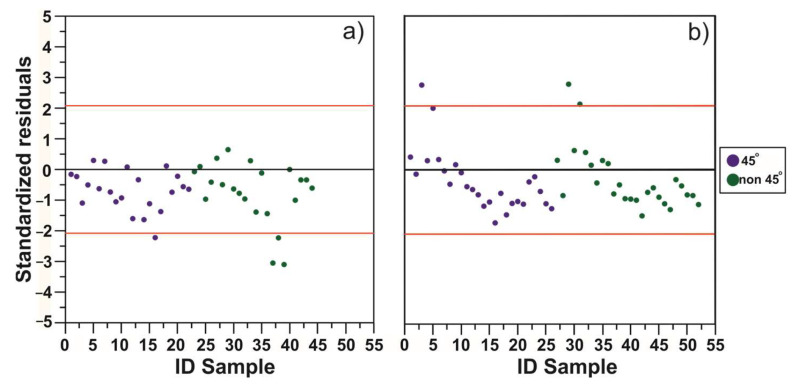
Analysis of standardized residuals. (**a**) $Z_{SD}$ model; (**b**) *C**h**l_a_* model.

**Table 1 sensors-23-03199-t001:** Number of stations from which data for each variable were obtained.

Cruise	Images	$Z_{SD}$	*E_d_* (*PAR*)	$a\left(\lambda \right)$	*C* *h* *l_a_*
Point sur 0413	15	11	11	9	15
Pangas 0613	29	27	27	26	29
Glyders 0615	15	12	12	11	14
Marías 0316	8	8	8	7	5
Vaquita 0716	7	4	4	7	7
Exfinife 0916	7	7	7	7	7
Marías 0916	5	4	4	2	5
**TOTAL**	**86**	**73**	**73**	**69**	**82**

**Table 2 sensors-23-03199-t002:** Chromaticity coordinates (*x*, *y*) estimated by Novoa et al. [60] for each Forel–Ule water type.

Forel–Ule Scale	*x*	*y*	Forel–Ule Scale	*x*	*y*	Forel–Ule Scale	*x*	*y*
**1**	0.191	0.167	8	0.315	0.440	15	0.446	0.458
**2**	0.199	0.200	9	0.337	0.462	16	0.461	0.449
**3**	0.210	0.240	10	0.363	0.476	17	0.475	0.441
**4**	0.227	0.288	11	0.386	0.487	18	0.489	0.433
**5**	0.246	0.335	12	0.402	0.481	19	0.503	0.425
**6**	0.266	0.376	13	0.416	0.474	20	0.516	0.416
**7**	0.297	0.412	14	0.431	0.466	21	0.528	0.408

**Table 3 sensors-23-03199-t003:** Results of the stepwise multiple regression analysis of the models for the general approach (α=0.05).

Variable	n	Model	Min	Max	R $\left(t_{cal}\right)$	G $\left(t_{cal}\right)$	B $\left(t_{cal}\right)$	$t_{cri}$	$F_{cal}$	$F_{cri}$	$R^{2}$
Forel–Ule scale	36	=3.16 + 0.0188 R+ 0.0495 G −0.0335 B (Equation (18))	1	8	2.20	22.47	−12.08	1.99	242.42	3.12	96%
$Z_{SD}$	30	= 22.1 − 0.309 G + 0.201 B (Equation (19))	4 m	35 m		−14.45	5.13	2.00	109.13	4.00	89%
$K_{d\;}$	35	= 0.218 + 0.00179 G − 0.00196 B (Equation (20))	0.045 $m^{-1}$	0.293$m^{-1}$		7.79	−5.78	1.99	30.57	3.98	66%
Jerlov scale	28	= 37 – 0.0733 G – 0.0874 B (Equation (21))	IA	4C		−5.56	−4.45	2.00	83.58	4.02	87%
$Chl_{a}$	31	= 0.521 + 0.00534 G − 0.00521 B (Equation (22))	0.053 $\frac{\mathrm{mg}}{m^{3}}$	0.541 $\frac{\mathrm{mg}}{m^{3}}$		11.19	−7.73	2.00	62.83	4.00	82%

**Table 4 sensors-23-03199-t004:** Results of the validation analysis of the models for the general approach.

Variable	Model	n	$r_{p\;cal}$	$r_{p\;cri}$
Forel–Ule scale	= 3.16 + 0.0188 R + 0.0495 G − 0.0335 B (Equation (18))	36	0.96	0.33
$Z_{SD}$	= 22.1 − 0.309 G + 0.201 B (Equation (19))	30	0.81	0.36
$K_{d\;}$	= 0.218 + 0.00179 G − 0.00196 B (Equation (20))	34	0.69	0.33
Alternative $K_{d\;}$	Apply Equation (15) to obtain $Z_{SD}$; then, apply criteria from Castillo-Ramírez et al. [56]	30	0.85	0.36
Jerlov scale	= 37 − 0.0733 G − 0.0874 B (Equation (21))	28	0.90	0.37
$Chl_{a}$	= 0.521 + 0.00534 G − 0.00521 B (Equation (22))	31	0.76	0.36

**Table 5 sensors-23-03199-t005:** Results of stepwise multiple regression analysis for the oceanic approach models (α=0.05).

Variable	*n*	Model	R $\left(t_{cal}\right)$	G $\left(t_{cal}\right)$	B $\left(t_{cal}\right)$	$t_{crit}$	$F_{cal}$	$F_{crit}$
Forel–Ule scale	16	= 3.16 + 0.0221 R + 0.0495 G − 0.0346 B (Equation (23))	2.43	12.49	−4.80	2.04	71.74	3.34
$Z_{SD}$	12			2.06		2.08	4.24	4.30
$K_{d\;}$	14		0.99	1.64	0.96	2.05	1.64	3.38
Jerlov scale	16		1.29	0.07	0.77	2.04	0.91	3.34
$Chl_{a}$	14			1.69	−0.51	2.07	1.45	4.32

**Table 6 sensors-23-03199-t006:** Results of the stepwise multiple regression analysis for the coastal approach models (α=0.05).

Variable	n	Model	Min	Max	R $\left(t_{cal}\right)$	G $\left(t_{cal}\right)$	B $\left(t_{cal}\right)$	$t_{cri}$	$F_{cal}$	$F_{cri}$	$R^{2}$
Forel–Ule scale	18	= 5.12 + 0.0406 G − 0.0368 B (Equation (24))	3	6		9.16	−13.16	2.03	161.32	4.14	92%
$Z_{SD}$	13	= 29.2 − 0.369 G + 0.210 B (Equation (25))	4 m	30 m		−7.46	6.26	2.07	63.66	3.44	88%
$K_{d\;}$	12	= −0.191 + 0.00329 R + 0.0018 G (Equation (26))	0.053 $m^{-1}$	0.127$m^{-1}$	5.08	5.06		2.08	21.40	4.37	73%
Jerlov scale	16	= 2.92 + 0.0538 G (Equation (27))	1C	4C		5.49		2.04	30.13	3.32	55%
$Chl_{a}$	16	= 1.44 + 0.00243 G − 0.00808 B (Equation (28))	0.080 $\frac{\mathrm{mg}}{m^{3}}$	0.703 $\frac{\mathrm{mg}}{m^{3}}$		2.82	−8.98	2.04	56.66	4.18	82%

**Table 7 sensors-23-03199-t007:** Results of the model validation analysis for the oceanic and coastal approaches.

Variable	Model	n	$r_{P\;cal}$	$r_{P\;cri}$
Forel–Ule scale	= 3.16 + 0.0221 R + 0.0495 G − 0.0346 B (Equation (23))	15	0.93	0.51
Forel–Ule scale	= 5.12 + 0.0406 G − 0.0368 B (Equation (24))	17	0.95	0.48
$Z_{SD}$	= 29.2 − 0.369 G + 0.210 B (Equation (25))	12	0.90	0.57
$K_{d\;}$	= −0.191 + 0.00329 R + 0.0018 G (Equation (26))	11	0.80	0.60
Coastal alternative $K_{d\;}$	Apply Equation (21) to obtain $Z_{SD}$; then, apply criteria from Castillo-Ramírez et al. [56]	12	0.97	0.57
Jerlov scale	= 2.92 + 0.0538 G (Equation (27))	16	0.73	0.49
$Chl_{a}$	= 1.44 + 0.00243 G − 0.00808 B (Equation (28))	16	0.85	0.49

**Table 8 sensors-23-03199-t008:** $Chl_{a}$ general and coastal models compared with literature models.

General Approach							Coastal Approach						
Model	n	$R^{2}$	RMSE	BIAS	MAE	MPI	Model	n	$R^{2}$	RMSE	BIAS	MAE	MPI
$Chl_{a}$(Equation (22))	31	82%	0.075	0.013	0.063	0.66	$Chl_{a}$(Equation (28))	16	82%	0.082	0.027	0.071	0.66
CP885 [48]	31	49%	0.085	0.112	0.115	0.33	CP885 [48]	16	49%	0.107	0.158	0.162	0.33
ECOShot [48]	31	53%	0.104	0.209	0.209	0.00	ECOShot [48]	16	53%	0.139	0.311	0.311	0.00

**Table 9 sensors-23-03199-t009:** Results of the non-parametric Friedman analysis (α = 0.05), where $F_{r\;crit}\;$ = $\chi_{\left(\alpha ,k-1\right)}^{2}$.

	*Z_SD_*	*C* *h* *l_a_*
**n**	22	26
$F_{r\;cal}$	11.02	11.07
$F_{r\;crit}$	5.99	5.99

**Table 10 sensors-23-03199-t010:** Least-squares analysis for the and $Z_{SD}$ and *C**h**l_a_* models.

	$Z_{SD}$	*C* *h* *l_a_*
**n**	22	26
$X_{45^{o}}^{2}$	52	5.82
$X_{non-45^{o}}^{2}$	55	11.85

## Data Availability

Not applicable.

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
