# Peer review of "Use of Digital Images as a Low-Cost System to Estimate Surface Optical Parameters in the Ocean"

_sensors, 2023, doi:10.3390/s23063199_

Round 1

Reviewer 1 Report

- The numerical results obtained can be reflected in the abstract.

- The authors have to clearly discuss about the main motivations and contributions of this study.

- A section, recent works can be added to summarize recent works such as the following Ch, A., Ch, R., Gadamsetty, S., Iwendi, C., Gadekallu, T. R., & Dhaou, I. B. (2022). ECDSA-based water bodies prediction from satellite images with UNet. Water14(14), 2234.

- The authors have to present a detailed discussion on the results obtained that include the inferences on the better performance of the proposed model.

- Compare the results obtained with recent state of the art.

- Analyse the computational complexity of the proposed approach.

- What are the threats to validity of the proposed approach?

Author Response

General comments by Reviewer 1.

 The numerical results obtained can be reflected in the abstract.

Thanks for your observation. We agree that this information should be added. In the original manuscript we did not include because “Sensors” author instructions said to keep the abstract limited to 200 words. According to your observation we have added numerical results trying to extend the abstract the minimum possible.

These are the new sentences in the abstract (lines 21-25) and the total word count is 222:

The results of the coastal approach showed higher correlations between the modeled and validation data, with  values of 0.80 for , 0.90 for , 0.85 for ?ℎ??, 0.73 for Jerlov, and 0.95 for Forel-Ule. Besides, the oceanic approach failed to detect significant changes in a digital photograph. The more precise results were obtained when images were captured at 45˚ (n=22; ).

  1. The authors have to clearly discuss about the main motivations and contributions of this study.

We increase the discussion following your considerations and we add some text about this point in lines 513-557:

The results of this study have shown the potential of digital images to evaluate the surface optical parameters of a water plot. The main advantage of the proposed approaches is their easy implementation and low cost since it does not demand optics expertise and only requires a digital camera. These advantages can be summarized as follows [47-54,91-96]:

  1. Low-cost system: The use of smartphones, tablets, or digital cameras for capturing digital images is a cost-effective method for estimating surface optical parameters in the ocean, as it does not require expensive specialized equipment. Smartphones and other devices are relatively inexpensive and widely available, making them an attractive option for researchers who are working with limited budgets or do not have access to specialized instrumentation.
  2. Accessibility and easy use: Since these electronic devices are widely used, this methodology can provide a widespread network of data acquisition, which can be essential for glob-al-scale analyses and modeling.
  3. High spatial and temporal resolution: Digital images capturing can provide high spatial and temporal resolution, allowing for a more detailed analysis of the ocean's surface optical properties over a larger area.
  4. Versatility: Smartphones, tablets or digital cameras can be used to capture images from different platforms such as beaches, piers, and boats, making it a versatile tool for surface optical parameter estimation.
  5. Citizen Science: The use of digital images allows for citizen science to contribute to the data acquisition process. This can increase public participation in scientific research and environmental monitoring and their awareness and engagement in oceanographic research.
  6. Environmental monitoring: Surface optical parameters play a crucial role in the health of marine ecosystems. The use of smartphones can aid in environmental monitoring efforts, providing critical information for decision-making and conservation efforts.
  7. Rapid response: In the event of phytoplankton bloom, the use of digital im-ages can aid in rapid response efforts, providing real-time information on the extent and severity of the spill.

This type of low-cost, user-friendly approaches would not only benefit scientists, but could also be used by the tourism or aquaculture industries, as well as by individual citizens concerned about changes in the water quality in their local environment. The benefits of this approach go beyond just improving data collection. By enabling re-al-time and continuous monitoring of surface optical parameters, this approach provides valuable information for environmental management and decision-making. The enhanced resolution and coverage of the data also offer new opportunities for researchers to investigate the complex relationships between surface optical parameters and environmental processes. This is especially important in coastal areas, where our results are more accurate, since 355 million people are expected to live within the 100 km of the coast between 2020 and 2035 [16,97,98]. This continuously increasing pressure may probably lead to the degradation of these ecosystems, adversely impacting the ESs provided [26,99]. We propose implementing the use of the coastal approach to supplement the analyses traditionally used for in-situ monitoring.

  1. A section, recent works can be added to summarize recent works such as the following Ch, A., Ch, R., Gadamsetty, S., Iwendi, C., Gadekallu, T. R., & Dhaou, I. B. (2022). ECDSA-based water bodies prediction from satellite images with UNet. Water14(14), 2234.

We added a section in lines: 87-120:

The ability to obtain RGB digital color intensities from digital camera images taken from fixed platforms, boats or unmanned aerial vehicles has been evaluated in other studies [47-54]. The objective of these studies was to obtain water leaving radiance [54] or RGB reflectances to estimate surface optical parameters [48-51,53]. These latter approaches are based on the mixing ratio of each color ranges from 0 to 255, where zero indicates the absence of color and 255 the maximum mixing ratio. Based on this, each color is defined according to three numbers or digital values (R, G, B) [48-51,53, 54]. Once digital values have been obtained, they are associated with in-situ optical parameters of the surface studied [48-51,53, 54].

For marine waters Goddijn-Murphy et al. [48] developed two approaches to estimate surface chlorophyll a . These authors used conventional digital cameras, the Nikon Coolpix885(®) and the SeaLife ECOshot(®) to perform measurements at 12 stations located in Galway Bay (North Atlantic). They used the Nikon Coolpix885 (®) with a plastic tube fitted around the lens to break the air-water interface, to prevent surface reflected light from entering the camera and to allow the camera to capture only water leaving radiance; (2) They used the SeaLife ECOshot(®), a waterproof digital camera, to operate below the water surface, taking pictures of upwelling light, and then they inverted it to capture downwelling sunlight. In this work they obtained a significant but low relationship between the colors B and G of the images and the concentration of .

Based on the previous results and the development of smartphones, the cameras integrated into these devices began to be used to remotely monitor the optical characteristics of a water parcel using mobile applications (app), such as:

- Secchi3000 app was designed to estimate turbidity and ZSD. Along with the app, a simple and cheap device is provided that must be filled with the water to be tested, and then a photograph must be taken in specific areas of the sample. The image is sent to the server, where it is processed using pattern recognition and computer vision techniques [49].

- TheEyeofWater app allows evaluating the water color. The app guides the user to capture an image of the water surface. After capturing the image, the observed water color is assigned one of the 21 colors on the Forel-Ule scale [50].

-HydroColor app allows taking images of the water surface following a specific protocol that includes the positioning angle of the camera and the observer. Subsequently, the app processes the colors of the image to estimate the turbidity of the water [51].

  1. The authors have to present a detailed discussion on the results obtained that include the inferences on the better performance of the proposed model.

We increase the discussion following your considerations and we add some text about this point in lines 343 and 360

The approaches proposed in this work are empirical therefore, they depend on the boundary limits established by the variables used in their development [26, 74]. Empirical modeling of ocean color and optical parameters is challenged by limited sampling opportunity on sunny days with calm seas. These conditions are essential to obtain data that rep-resents the true variability of the light field in water [75-77]. To address this challenge, this research used data from seven oceanographic cruises carried out between 2013 and 2016. In each cruise, a sampling network of more than 70 stations was established. However, data could only be obtained at approximately 15% of the established stations due to unfavorable conditions (e. g., cloudy days, waves) (Table 1). In addition, it is necessary to obtain an independent data set to validate the developed model. This validation is essential to assess the ability of the model to predict values [26, 74].

To address these challenges, the database for each variable was divided into two independent groups, using 50% of the data for the modeling process and the other 50% for validation. In addition, to ensure the robustness of the models, five iterations were performed randomly selecting five different data sets to model, each with its corresponding validation set [56, 69, 70] (Fig.4). This approach helped to reduce the random error associated with the selection of the two data sets.

Corrected Figure 4. Scheme of approaches for each optical parameter and the database used for its development.

  1. Compare the results obtained with recent state of the art.

We increase the discussion following your considerations and we add some text about this point in lines 295-320, 450-472:

In order to compare the models estimated in this work with those from the literature, three statistical descriptors were calculated: mean absolute error ( ) (Eq. 13), root-mean-square error ( ) (Eq. 14), and analysis of bias (BIAS) (Eq. 15).

                     (13)

                  (14)

where n is the total number of data included in this analysis,   is residual observations, and  is the absolute value of residuals.

      (15)

where BIAS is the residual mean.

According to  and , lower values represent better results, whereas BIAS values closer to zero mean better results. To determine which was the best model, the model performance index (MPI) was estimated [56] (Eq. 16), which is based on the three statistical descriptors mentioned above.

(16)

Where  are the range of  y , respectively.  is the absolute range of , and p is the total number of compared models. The ranks were calculated following the criteria of Wilcoxon [72]. The MPI intervals range from 0 to 1, where values closer to 1 represent the better model.

In table 8 we compared the performance of our  general and coastal models (Eq. 22 y 28) with Goddijn-Murphy et al. [48] models. These authors developed two models, in which the B/G ratios are the independent variables and is the dependent variable. 

         Tabla 8.  general and coastal models compared with literature models.

General approach

Coastal approach

Model

n

RMSE

BIAS

MAE

MPI

Model

n

RMSE

BIAS

MAE

MPI

(Eq. 22)

31

82%

0.075

0.013

0.063

0.66

(Eq. 28)

16

82%

0.082

0.027

0.071

0.66

CP885 [48]

31

49%

0.085

0.112

0.115

0.33

CP885 [48]

16

49%

0.107

0.158

0.162

0.33

ECOShot [48]

31

53%

0.104

0.209

0.209

0.00

ECOShot [48]

16

53%

0.139

0.311

0.311

0.00

The results of table 8 showed that the models proposed in this work had a better performance in the validation process compared to those reported in the literature applied to our data. These show the relevance to acquire field data under appropriate conditions (e.g., sunny days, calm sea). These conditions were not observed in Goddijn-Murphy et al. [48]. Also, the lower R2 obtained by Goddijn-Murphy et al. [48] could be due to the high CDOM concentrations reported, which could affect the absorption of blue wavelengths in water.

Gao et al. [53] developed an algorithm to estimate  using smartphone images in continental water bodies. These authors observed the same conditions that in our study, including camera and sun angles. However, we could not apply their model to our data because it was developed for other conditions. Continental waters with high CDOM concentrations are characterized by high reflectance in R and high absorption in B [48]. Besides, they used a limnologic Secchi disk, while in this research an oceanic disk was used. Differences in terms of size and reflectance surface are noticeable in both versions [26,56], so they are not comparable.

  1. Analyze the computational complexity of the proposed approach.

We have answered this question together with question 4, revised manuscript lines 343 and 360

  1. What are the threats to validity of the proposed approach?

We increased the discussion following your considerations and we added some text about this point in revised manuscript lines 599-629, also having into account reviewer 2 suggestions.:

Although the use of digital images to estimate surface optical parameters in the ocean has notable advantages in terms of ease of use and low cost, it is important to mention that there are limitations that must be considered. One issue that could arise is whether different technologies used in smartphone cameras could generate different RGB readings. However, Leeuw and Boss [51] evaluated the spectral sensitivity of RGB channels in different next generation devices, and showed that, although there may be differences in the spectral shape, the values in RGB peaks are practically the same between devices. The approaches presented in this work were based on the RGB peaks (Fig. 3), so they should be valid regardless of the device being used. Another issue is the interpretation of color changes in a relatively small area (1 cm x 1cm). Our methodology was based on the work of Leeuw and Boss [51] who proposed establishing a fixed area within the photograph so that devices with cameras of different resolutions could be compared with each other, thanks to the fact that the field of view of the camera it would be the same between devices.

            Variability in lighting and image quality can be critical factors that influence the quality of the data obtained, as mentioned by [75-77]. The quantity and quality of light reaching the ocean surface depends mainly on the position of the sun and cloud cover [27,28]. The image quality can be affected by elements influencing the visual state of the ocean surface, for example, the presence of waves or white caps, the sunshine, or the shadows generated by the boat or platform where the image is captured [50,51,53,58,59].

To overcome these limitations and obtain quality data, we followed the reflectance measurement methodology used with the SIMBAD spectroradiometer to capture the photographs (refer to [58]). As mentioned by Fougnie et al. [108] and Deschamps et al. [58], this methodology is very specific about the sun position angle and the equipment angles, and it allows to reduce the noise or interference caused by the sunshine and the reflection of the sky on the water surface. Digital images were captured at an angle of 45˚ to the ocean surface on sunny days, when the sun was at an angle equal to or greater than 45˚ to the horizon, and with little or no cloud cover. The 45˚ sun position at the zenith minimizes Fresnel reflectance at the water surface allowing greater light penetration, improving the accuracy of radiometric measurements, and facilitating accurate estimation of optical parameters in the ocean [108].

Reviewer 2 Report

A use of simple, cheap and portable devices to monitor the ambient environment worldwide is an attractive complement (and counterpart) to sophisticated and experimentally complex field measurements. On the one hand, the methods based on digital camera technology are widely implemented in many fields of science. On the other hand, there is a number of limitations that need to be taken into account.

The title and motivation of the present paper are promising. However, I have serious doubts about the solution based on cell phone cameras since different technologies can result in different image processing and even RGB readouts. In the present paper the authors interpret color changes in a relatively small color space, so a small difference in image processing among different smartphone technologies can potentially results in large errors. This is an essential shortcoming for this work.

In addition, the light field scattered underwater depends on geometry of direct sunbeam relative to the sensor viewing geometry (Sci Rep 4, 3748, 2014). Thus the intensity and spectrum would change with the scattering angle. Averaging of the values obtained from all quadrants in all the images selected for each station could be then a source of interpretation errors.

Why the zenith angle 45 deg has been used?

From where do the data in Tab.1 came out? No experiment for a(lambda),… is documented in the paper.

A validation test against data obtained independently is missing.

Author Response

General comments by Reviewer 2.

 A use of simple, cheap and portable devices to monitor the ambient environment worldwide is an attractive complement (and counterpart) to sophisticated and experimentally complex field measurements. On the one hand, the methods based on digital camera technology are widely implemented in many fields of science. On the other hand, there is a number of limitations that need to be taken into account.

We increase the discussion following your considerations and we add some text about this point in the revised manuscript lines 599-629, also having into account reviewer 1 suggestions:

Although the use of digital images to estimate surface optical parameters in the ocean has notable advantages in terms of ease of use and low cost, it is important to mention that there are limitations that must be considered. One issue that could arise is whether different technologies used in smartphone cameras could generate different RGB readings. However, Leeuw and Boss [51] evaluated the spectral sensitivity of RGB channels in different next generation devices, and showed that, although there may be differences in the spectral shape, the values in RGB peaks are practically the same between devices. The approaches presented in this work were based on the RGB peaks (Fig. 3), so they should be valid regardless of the device being used. Another issue is the interpretation of color changes in a relatively small area (1 cm x 1cm). Our methodology was based on the work of Leeuw and Boss [51] who proposed establishing a fixed area within the photograph so that devices with cameras of different resolutions could be compared with each other, thanks to the fact that the field of view of the camera it would be the same between devices.

            Variability in lighting and image quality can be critical factors that influence the quality of the data obtained, as mentioned by [75-77]. The quantity and quality of light reaching the ocean surface depends mainly on the position of the sun and cloud cover [27,28]. The image quality can be affected by elements influencing the visual state of the ocean surface, for example, the presence of waves or white caps, the sunshine, or the shadows generated by the boat or platform where the image is captured [50,51,53,58,59].

To overcome these limitations and obtain quality data, we followed the reflectance measurement methodology used with the SIMBAD spectroradiometer to capture the photographs (refer to [58]). As mentioned by Fougnie et al. [108] and Deschamps et al. [58], this methodology is very specific about the sun position angle and the equipment angles, and it allows to reduce the noise or interference caused by the sunshine and the reflection of the sky on the water surface. Digital images were captured at an angle of 45˚ to the ocean surface on sunny days, when the sun was at an angle equal to or greater than 45˚ to the horizon, and with little or no cloud cover. The 45˚ sun position at the zenith minimizes Fresnel reflectance at the water surface allowing greater light penetration, improving the accuracy of radiometric measurements, and facilitating accurate estimation of optical parameters in the ocean [108].

  1. The title and motivation of the present paper are promising. However, I have serious doubts about the solution based on cell phone cameras since different technologies can result in different image processing and even RGB readouts. In the present paper the authors interpret color changes in a relatively small color space, so a small difference in image processing among different smartphone technologies can potentially results in large errors. This is an essential shortcoming for this work.

Thanks for your observation, it is a good point and in fact it has also been tackled by other authors. We added a new reference in methodology section, line 180 and explained in discussion section, lines 601-612

One issue that could arise is whether different technologies used in smartphone cameras could generate different RGB readings. However, Leeuw and Boss [51] evaluated the spectral sensitivity of RGB channels in different next generation devices, and showed that, although there may be differences in the spectral shape, the values in RGB peaks are practically the same between devices. The approaches presented in this work were based on the RGB peaks (Fig. 3), so they should be valid regardless of the device being used. Another issue is the interpretation of color changes in a relatively small area (1 cm x 1cm). Our methodology was based on the work of Leeuw and Boss [51] who proposed establishing a fixed area within the photograph so that devices with cameras of different resolutions could be compared with each other, thanks to the fact that the field of view of the camera it would be the same between devices.

  1. In addition, the light field scattered underwater depends on geometry of direct sunbeam relative to the sensor viewing geometry (Sci Rep 4, 3748, 2014). Thus the intensity and spectrum would change with the scattering angle. Averaging of the values obtained from all quadrants in all the images selected for each station could be then a source of interpretation errors.

Thank you for your comment. First, due to a translation error, the word “average” was used instead of “mean”.

It is a good point that we had into account in our analysis as follows. In our approach, we used the mean of the values obtained from the four quadrants of each of the six photographs for each station, which allowed us to establish a representative midpoint of reference for the RGB peaks of the station to be evaluated, as described shows below (graph 1):

Graph 1. X axis indicates the RGB color, where B=1, R=2 and G=3, Y axis is the RGB peak value. The lines indicate the mean value for each channel.

To ensure that the mean was representative and that there were no high noise values, the standardized anomalies of the data from each station were calculated following the guidelines of Santamaría-Ángel et al. (2019) and the criteria of the Six Sigma analysis (Bass, 2007). Here it is established that high noise values are within +- 3 standard deviations. This was carried out routinely in each station and high noise values were not found in any of them, so these data are not presented in the original or in the revised manuscript.

Graph 2 (same station as graph 1) shows that there are no high noise data that could influence the calculation of the mean as suggested by the reviewer.

Graph 2. X axis indicates the RGB color, where B=1, R=2 and G=3, Y axis are the standardized anomalies. The lines represent the limit established by Six Sigma analysis to detect high noise values.

  1. Why the zenith angle 45 deg has been used?

This information was added in the revised manuscript lines 621-629

Due to the expertise of the group in reflectance measurements using the SIMBAD spectroradiometer, the photographs were taken when the sun was 45˚ above the horizon based on the publications by Fougnie et al. [92] and Deschamps et al. [54].

The 45˚ sun position at the zenith minimizes Fresnel reflectance at the water surface allowing greater light penetration, improving the accuracy of radiometric measurements, and facilitating accurate estimation of optical parameters in the ocean [108].

  1. From where do the data in Tab.1 came out? No experiment for a (lambda) is documented in the paper.

Table 1 shows the number of stations sampled per cruise for each variable. It is important to highlight that the number of stations varies according to the variable and the cruise, since the sampling depends to a great extent on the suitable conditions to measure the optical parameters. To obtain data that reflects the true variability of the light field in the water, sampling conditions of sunny days with calm seas and no shadows generated by the ship on the water surface are required.

It should be noted that the data of the total light absorption coefficient (a(λ)) were used as an intermediate step to classify the stations according to the Jerlov scale, following the protocols described by Solonenko and Mobley (2015) and Castillo-Ramírez et al. (2020). This is described in the manuscript in the methodology section in the lines 221-223.

  1. A validation test against data obtained independently is missing.

Model validation was carried out using an independent data set, maybe it was not clearly enough explained in the original manuscript. To clarify this point, figure 4 in the revised manuscript was modified to include a section dedicated to validation. The results of the validation are presented in Tables 4 and 7. In addition, considering also the observations of reviewer 1, an explanation on this point was included in discussion lines 343 and 360:

The approaches proposed in this work are empirical, they depend on the boundary limits established by the variables used in their development [26, 74]. Empirical modeling of ocean color and optical parameters is challenged by limited sampling opportunity on sunny days with calm seas. These conditions are essential to obtain data that rep-resents the true variability of the light field in water [75-77]. To address this challenge, this research used data from seven oceanographic cruises carried out between 2013 and 2016. In each cruise, a sampling network of more than 70 stations was established. However, data could only be obtained at approximately 15% of the established stations due to unfavorable conditions (e. g., cloudy days, waves) (Table 1). In addition, it is necessary to obtain an independent data set to validate the developed model. This validation is essential to assess the ability of the model to predict values [26, 74].

To address these challenges, the database for each variable was divided into two independent groups, using 50% of the data for the modeling process and the other 50% for validation. In addition, to ensure the robustness of the models, five iterations were performed randomly selecting five different data sets to model, each with its corresponding validation set [56, 69, 70] (Fig.4). This approach helped to reduce the random error associated with the selection of the two data sets. 

Corrected Figure 4. Scheme of approaches for each optical parameter and the database used for its development.

Round 2

Reviewer 1 Report

ll the comments are addressed

Reviewer 2 Report

Although I am still not fully convinced with the capabilities of the method as it is presented, I am inclined to accept the paper as is and let a wider scientific community to comment on it after being published.